# PHD: Pixel-Based Language Modeling of Historical Documents

**Nadav Borenstein   Phillip Rust   Desmond Elliott   Isabelle Augenstein**
Department of Computer Science, University of Copenhagen
{nadav.borenstein, p.rust, de, augenstein}@di.ku.dk

## Abstract

The digitisation of historical documents has provided historians with unprecedented research opportunities. Yet, the conventional approach to analysing historical documents involves converting them from images to text using OCR, a process that overlooks the potential benefits of treating them as images and introduces high levels of noise. To bridge this gap, we take advantage of recent advancements in pixel-based language models trained to reconstruct masked patches of pixels instead of predicting token distributions. Due to the scarcity of real historical scans, we propose a novel method for generating synthetic scans to resemble real historical documents. We then pre-train our model, PHD, on a combination of synthetic scans and real historical newspapers from the 1700-1900 period. Through our experiments, we demonstrate that PHD exhibits high proficiency in reconstructing masked image patches and provide evidence of our model's noteworthy language understanding capabilities. Notably, we successfully apply our model to a historical QA task, highlighting its utility in this domain.

## 1   Introduction

Recent years have seen a boom in efforts to digitise historical documents in numerous languages and sources (Chadwyck, 1998; Groesen, 2015; Moss, 2009), leading to a transformation in the way historians work. Researchers are now able to expedite the analysis process of vast historical corpora using NLP tools, thereby enabling them to focus on interpretation instead of the arduous task of evidence collection (Laite, 2020; Gerritsen, 2012).

The primary step in most NLP tools tailored for historical analysis involves Optical Character Recognition (OCR). However, this approach poses several challenges and drawbacks. First, OCR strips away any valuable contextual meaning embedded within non-textual elements, such as page layout, fonts, and figures.[1] Moreover, historical documents present numerous challenges to OCR systems. This can range from deteriorated pages, archaic fonts and language, the presence of non-textual elements, and occasional deficiencies in scan quality (e.g., blurriness), all of which contribute to the introduction of additional noise. Consequently, the extracted text is often riddled with errors at the character level (Robertson and Goldwater, 2018; Bollmann, 2019), which most large language models (LLMs) are not tuned to process. Token-based LLMs are especially sensitive to this, as the discrete structure of their input space cannot handle well the abundance of out-of-vocabulary words that characterise OCRed historical documents (Rust et al., 2023). Therefore, while LLMs have proven remarkably successful in modern domains, their performance is considerably weaker when applied to historical texts (Manjavacas and Fonteyn, 2022; Baptiste et al., 2021, *inter alia*). Finally, for many languages, OCR systems either do not exist or perform particularly poorly. As training new OCR models is laborious and expensive (Li et al., 2021a), the application of NLP tools to historical documents in these languages is limited.

This work addresses these limitations by taking advantage of recent advancements in pixel-based language modelling, with the goal of constructing a general-purpose, image-based and OCR-free language encoder of historical documents. Specifically, we adapt PIXEL (Rust et al., 2023), a language model that renders text as images and is trained to reconstruct masked patches instead of predicting a distribution over tokens. PIXEL's training methodology is highly suitable for the historical domain, as (unlike other pixel-based language models) it does not rely on a pretraining dataset

---

*This paper shows dataset samples that are racist in nature

[1]Consider, for example, the visual data that is lost by processing the newspaper page in Fig 18 in App C as text.

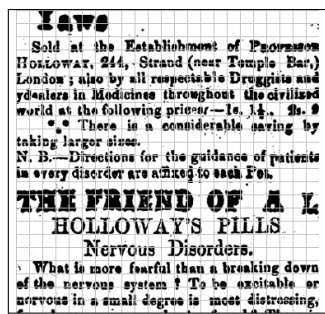

(a) Input example.

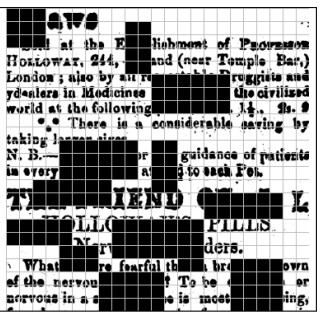

(b) Masking the input.

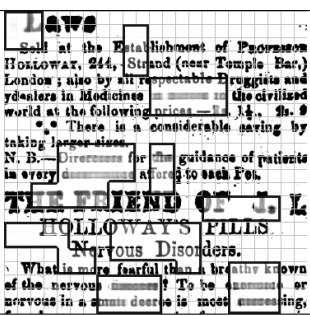

(c) Model predictions.

Figure 1: Our proposed model, PHD. The model is trained to reconstruct the original image (a) from the masked image (b), resulting in (c). The grid represents the 16 × 16 pixels patches that the inputs are broken into.

composed of instances where the image and text are aligned. Fig 1 visualises our proposed training approach.

Given the paucity of large, high-quality datasets comprising historical scans, we pretrain our model using a combination of 1) synthetic scans designed to resemble historical documents faithfully, produced using a novel method we propose for synthetic scan generation; and 2) real historical English newspapers published in the Caribbeans in the 18th and 19th centuries. The resulting pixel-based language encoder, PHD (**P**ixel-based model for **H**istorical **D**ocuments), is subsequently evaluated based on its comprehension of natural language and its effectiveness in performing Question Answering from historical documents.

We discover that PHD displays impressive reconstruction capabilities, being able to correctly predict both the form and content of masked patches of historical newspapers (§4.4). We also note the challenges concerning quantitatively evaluating these predictions. We provide evidence of our model's noteworthy language understanding capabilities while exhibiting an impressive resilience to noise. Finally, we demonstrate the usefulness of the model when applied to the historical QA task (§5.4).

To facilitate future research, we provide the dataset, models, and code at ⟳ https://github.com/nadavborenstein/pixel-bw.

## 2 Background

### 2.1 NLP for Historical Texts

Considerable efforts have been invested in improving both OCR accuracy (Li et al., 2021a; Smith, 2023) and text normalisation techniques for historical documents (Drobac et al., 2017; Robertson and Goldwater, 2018; Bollmann et al., 2018; Boll-

mann, 2019; Lyu et al., 2021). This has been done with the aim of aligning historical texts with their modern counterparts. However, these methods are not without flaws (Robertson and Goldwater, 2018; Bollmann, 2019), and any errors introduced during these preprocessing stages can propagate to downstream tasks (Robertson and Goldwater, 2018; Hill and Hengchen, 2019). As a result, historical texts remain a persistently challenging domain for NLP research (Lai et al., 2021; De Toni et al., 2022; Borenstein et al., 2023b). Here, we propose a novel approach to overcome the challenges associated with OCR in historical material, by employing an image-based language model capable of directly processing historical document scans and effectively bypassing the OCR stage.

### 2.2 Pixel-based Models for NLU

Extensive research has been conducted on models for processing text embedded in images. Most existing approaches incorporate OCR systems as an integral part of their inference pipeline (Appalaraju et al., 2021; Li et al., 2021b; Delteil et al., 2022). These approaches employ multimodal architectures where the input consists of both the image and the output generated by an OCR system.

Recent years have also witnessed the emergence of OCR-free approaches for pixel-based language understanding. Kim et al. (2022) introduce Donut, an image-encoder-text-decoder model for document comprehension. Donut is pretrained with the objective of extracting text from scans, a task they refer to as "pseudo-OCR". Subsequently, it is fine-tuned on various text generation tasks, reminiscent of T5 (Roberts et al., 2020). While architecturally similar to Donut, Dessurt (Davis et al., 2023) and Pix2Struct (Lee et al., 2022) were pretrained by masking image regions and predicting the text in

both masked and unmasked image regions. Unlike our method, all above-mentioned models predict in the text space rather than the pixel space. This presupposes access to a pretraining dataset comprised of instances where the image and text are aligned. However, this assumption cannot hold for historical NLP since OCR-independent ground truth text for historical scans is, in many times, unprocurable and cannot be used for training purposes.

Text-free models that operate at the pixel level for language understanding are relatively uncommon. One notable exception is Li et al. (2022), which utilises Masked Image Modeling for pretraining on document patches. Nevertheless, their focus lies primarily on tasks that do not necessitate robust language understanding, such as table detection, document classification, and layout analysis. PIXEL (Rust et al., 2023), conversely, is a text-free pixel-based language model that exhibits strong language understanding capabilities, making it the ideal choice for our research. The subsequent section will delve into a more detailed discussion of PIXEL and how we adapt it to our task.

## 3 Model

**PIXEL** We base PHD on PIXEL, a pretrained pixel-based encoder of language. PIXEL has three main components: A text renderer that draws texts as images, a pixel-based encoder, and a pixel-based decoder. The training of PIXEL is analogous to BERT (Devlin et al., 2019). During pretraining, input strings are rendered as images, and the encoder and the decoder are trained jointly to reconstruct randomly masked image regions from the unmasked context. During finetuning, the decoder is replaced with a suitable classification head, and no masking is performed. The encoder and decoder are based on the ViT-MAE architecture (He et al., 2022) and work at the patch level. That is, the encoder breaks the input image into patches of $16 \times 16$ pixels and outputs an embedding for each patch. The decoder then decodes these patch embeddings back into pixels. Therefore, random masking is performed at the patch level as well.

**PHD** We follow the same approach as PIXEL's pretraining and finetuning schemes. However, PIXEL's intended use is to process texts, not natural images. That is, the expected input to PIXEL is a string, not an image file. In contrast, we aim to use the model to encode real document scans. Therefore, we make several adaptations to PIXEL's

| Source | #Issues | #Train Scans | #Test Scans |
|---|---|---|---|
| Caribbean Project | 7 487 | 1 675 172 | 87 721 |
| Danish Royal Library | 5 661 | 300 780 | 15 159 |
| Total | 13 148 | 1 975 952 | 102 880 |

Table 1: Statistics of the newspapers dataset.

training and data processing procedures to make it compatible with our use case (§4 and §5).

Most crucially, we alter the dimensions of the model's input: The text renderer of PIXEL renders strings as a long and narrow image with a resolution of $16 \times 8464$ pixels (corresponding to $1 \times 529$ patches), such that the resulting image resembles a ribbon with text. Each input character is set to be not taller than 16 pixels and occupies roughly one patch. However, real document scans cannot be represented this way, as they have a natural two-dimensional structure and irregular fonts, as Fig 1a demonstrates (and compare to Fig 17a in App C). Therefore, we set the input size of PHD to be 368 $\times$ 368 pixels (or $23 \times 23$ patches).

## 4 Training a Pixel-Based Historical LM

We design PHD to serve as a general-purpose, pixel-based language encoder of historical documents. Ideally, PHD should be pretrained on a large dataset of scanned documents from various historical periods and different locations. However, large, high-quality datasets of historical scans are not easily obtainable. Therefore, we propose a novel method for generating historical-looking artificial data from modern corpora (see subsection 4.1). We adapt our model to the historical domain by continuously pretraining it on a medium-sized corpus of real historical documents. Below, we describe the datasets and the pretraining process of the model.

### 4.1 Artificially Generated Pretraining Data

Our pretraining dataset consists of artificially generated scans of texts from the same sources that BERT used, namely the BookCorpus (Zhu et al., 2015) and the English Wikipedia.[2] We generate the scans as follows.

We generate dataset samples on-the-fly, adopting a similar approach as Davis et al. (2023). First,

---

[2] We use the version "20220301.en" hosted on `huggingf ace.co/datasets/wikipedia`.

(a) Embedding one paragraph.    (b) Adding more paragraphs.    (c) Adding noise.

Figure 2: Process of generating a single artificial scan. Refer to §4.1 for detailed explanations.

we split the text corpora into paragraphs, using the new-line character as a delimiter. From a paragraph chosen at random, we pick a random spot and keep the text spanning from that spot to the paragraph's end. We also sample a random font and font size from a pre-defined list of fonts (from Davis et al. (2023)). The text span and the font are then embedded within an HTML template using the Python package Jinja,[3] set to generate a Web page with dimensions that match the input dimension of the model. Finally, we use the Python package WeasyPrint[4] to render the HTML file as a PNG image. Fig 2a visualises this process' outcome.

In some cases, if the text span is short or the selected font is small, the resulting image contains a large empty space (as in Fig 2a). When the empty space within an image exceeds 10%, a new image is generated to replace the vacant area. We create the new image by randomly choosing one of two options. In 80% of the cases, we retain the font of the original image and select the next paragraph. In 20% of the cases, a new paragraph and font are sampled. This pertains to the common case where a historical scan depicts a transition of context or font (e.g., Fig 1a). This process can repeat multiple times, resulting in images akin to Fig 2b.

Finally, to simulate the effects of scanning ageing historical documents, we degrade the image by adding various types of noise, such as blurring, rotations, salt-and-pepper noise and bleed-through effect (see Fig 2c and Fig 9 in App C for examples). App A.2 enumerates the full list of the degradations and augmentations we use.

## 4.2 Real Historical Scans

We adapt PHD to the historical domain by continuously pretraining it on a medium-sized corpus of

scans of real historical newspapers. Specifically, we collect newspapers written in English from the "Caribbean Newspapers, 1718–1876" database,[5] the largest collection of Caribbean newspapers from the 18th–19th century available online. We extend this dataset with English-Danish newspapers published between 1770–1850 in the Danish Caribbean colony of Santa Cruz (now Saint Croix) downloaded from the Danish Royal Library's website.[6] See Tab 1 for details of dataset sizes. While confined in its geographical and temporal context, this dataset offers a rich diversity in terms of content and format, rendering it an effective test bed for evaluating PHD.

Newspaper pages are converted into a $368 \times 368$ pixels crops using a sliding window approach over the page's columns. This process is described in more detail in App A.2. We reserve 5% of newspaper issues for validation, using the rest for training. See Fig 10 in App C for dataset examples.

## 4.3 Pretraining Procedure

Like PIXEL, the pretraining objective of PHD is to reconstruct the pixels in masked image patches. We randomly occlude 28% of the input patches with 2D rectangular masks. We uniformly sample their width and height from $[2, 6]$ and $[2, 4]$ patches, respectively, and then place them in random image locations (See Fig 1b for an example). Training hyperparameters can be found in App A.1.

## 4.4 Pretraining Results

**Qualitative Evaluation.** We begin by conducting a qualitative examination of the predictions made by our model. Fig 3 presents a visual representa-

---

[3] jinja.palletsprojects.com/en/3.1.x
[4] weasyprint.org
[5] readex.com/products/caribbean-newspapers-series-1-1718-1876-american-antiquarian-society
[6] statsbiblioteket.dk/mediestream

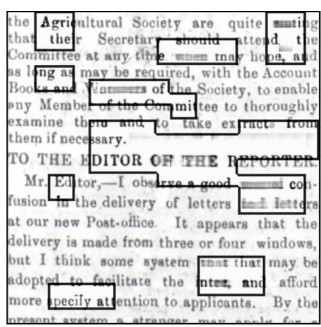 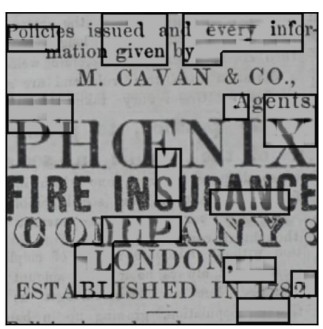 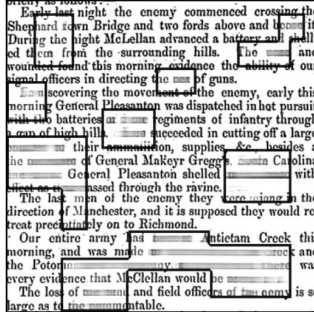

Figure 3: Examples of some image completions made by PHD . Masked regions marked by dark outlines.

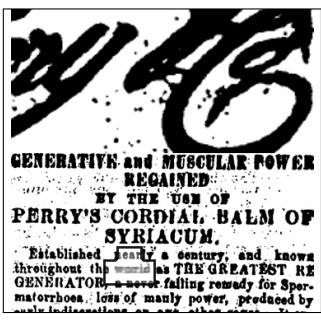 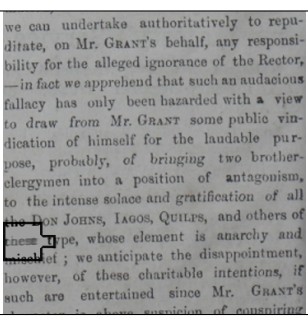

| (a) world | (b) 1893 | (c) every |

Figure 4: Single word completions made by our model. Figure captions depict the missing word. Fig (a) depicts a successful reconstruction, whereas Fig (b) and (c) represent fail-cases.

tion of the model's predictions on three randomly selected scans from the test set of the Caribbean newspapers dataset (for additional results on other datasets, refer to Fig 12 App C). From a visual inspection, it becomes evident that the model accurately reconstructs the fonts and structure of the masked regions. However, the situation is less clear when it comes to predicting textual content. Similar to Rust et al. (2023), unsurprisingly, prediction quality is high and the results are sharp for smaller masks and when words are only partially obscured. However, as the completions become longer, the text quality deteriorates, resulting in blurry text. It is important to note that evaluating these blurry completions presents a significant challenge. Unlike token-based models, where the presence of multiple words with high, similar likelihood can easily be detected by examining the discrete distribution, this becomes impossible with pixel-based models. In pixel-based completions, high-likelihood words may overlay and produce a blurry completion. Clear completions are only observed when a single word has a significantly higher probability compared to others. This limitation is an area that we leave for future work.

We now move to analyse PHD's ability to fill in single masked words. We randomly sample test

scans and OCRed them using Tesseract.[7] Next, we randomly select a single word from the OCRed text and use Tesseract's word-to-image location functionality to (heuristically) mask the word from the image. Results are presented in Fig 4. Similar to our earlier findings, the reconstruction quality of single-word completion varies. Some completions are sharp and precise, while others appear blurry. In some few cases, the model produces a sharp reconstruction of an incorrect word (Fig 4c). Unfortunately, due to the blurry nature of many of the results (regardless of their correctness), a quantitative analysis of these results (e.g., by OCRing the reconstructed patch and comparing it to the OCR output of the original patch) is unattainable.

**Semantic Search.** A possible useful application of PHD is semantic search. That is, searching in a corpus for historical documents that are semantically similar to a concept of interest. We now analyse PHD's ability to assign similar historical scans with similar embeddings. We start by taking a random sample of 1000 images from our test set and embed them by averaging the patch embeddings of the final layer of the model. We then reduce the dimensionality of the embeddings with

---

[7]github.com/tesseract-ocr/tesseract

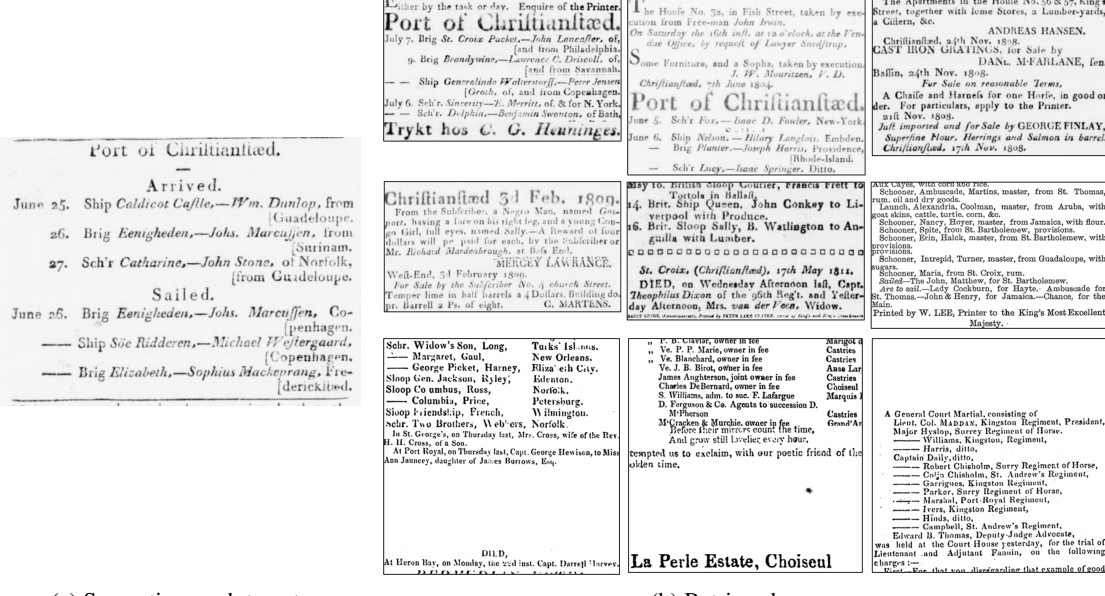

(a) Semantic search target.  (b) Retrieved scans.

Figure 5: Semantic search using our model. (a) is the target of the search, and (b) are scans retrieved from the newspaper corpus.

t-SNE (van der Maaten and Hinton, 2008). Upon visual inspection (Fig 13 in App C), we see that scans are clustered based on visual similarity and page structure.

Fig 13, however, does not provide insights regarding the semantic properties of the clusters. Therefore, we also directly use the model in semantic search settings. Specifically, we search our newspapers corpus for scans that are semantically similar to instances of the *Runaways Slaves in Britain* dataset, as well as scans containing shipping ads (See Fig 16 in App C for examples). To do so, we embed 1M random scans from the corpus. We then calculate the cosine similarity between these embeddings and the embedding of samples from the *Runaways Slaves in Britain* and embeddings of shipping ads. Finally, we manually examine the ten most similar scans to each sample.

Our results (Fig 5 and Fig 14 in App C) are encouraging, indicating that the embeddings capture not only structural and visual information, but also the semantic content of the scans. However, the results are still far from perfect, and many retrieved scans are not semantically similar to the search's target. It is highly plausible that additional specialised finetuning (e.g., SentenceBERT's (Reimers and Gurevych, 2019) training scheme) is necessary to produce more semantically meaningful embeddings.

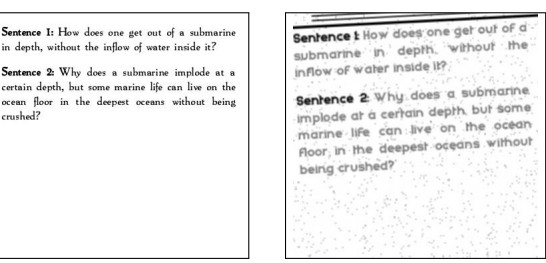

Figure 6: Samples from the clean and noisy visual GLUE datasets.

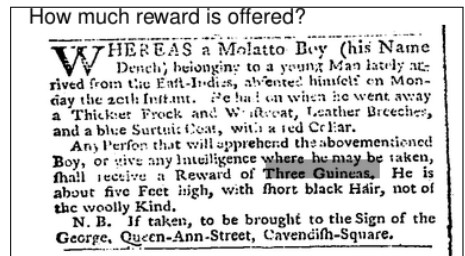

Figure 7: Example from the *Runaways Slaves in Britain* dataset, rendered as visual question answering task. The gray overlay marks the patches containing the answer.

## 5 Training for Downstream NLU Tasks

After obtaining a pretrained pixel-based language model adapted to the historical domain (§4), we now move to evaluate its understanding of natural language and its usefulness in addressing historically-oriented NLP tasks. Below, we describe the datasets we use for this and the experimental settings.

| Noise | Images | Model | MNLI 393k | QQP 364k | QNLI 105k | SST-2 67k | COLA 8.6k | STS-B 5.8k | MRPC 3.7k | RTE 2.5k | WNLI 635 | AVG |
|---|---|---|---|---|---|---|---|---|---|---|---|---|
| ✗ | ✗ | BERT | **84.1** | **87.6** | **91.0** | **92.6** | **60.3** | **88.8** | **90.2** | **69.5** | 51.8 | **80.0** |
|  |  | PIXEL | 78.5 | 84.5 | 87.8 | 89.6 | 38.4 | 81.1 | 88.2 | 60.5 | 53.8 | 74.1 |
|  | ✓ | CLIP$_{lin}$ | 50.2 | 64.7 | 67.4 | 79.8 | 4.2 | 56.4 | 74.1 | 51.5 | 25.6 | 52.7 |
|  |  | Donut | 64.0 | 77.8 | 69.7 | 82.1 | 13.9 | 14.4 | 81.7 | 54.0 | 57.7 | 57.2 |
|  |  | *Ours* | 70.1 | 82.7 | 82.3 | 82.5 | 15.9 | 80.2 | 83.4 | 59.9 | 54.1 | 67.9 |
| ✓ | ✓ | OCR+BERT | **71.7** | 77.5 | **82.7** | **85.5** | **39.7** | 68.4 | **86.9** | 58.8 | 51.3 | **69.2** |
|  |  | OCR+PIXEL | 70.6 | 78.5 | 81.5 | 83.6 | 30.3 | 68.8 | 84.7 | **59.7** | 58.6 | 68.5 |
|  |  | CLIP$_{lin}$ | 45.3 | 67.4 | 64.4 | 79.2 | 3.5 | 57.9 | 78.8 | 47.3 | 32.7 | 52.9 |
|  |  | Donut | 61.6 | 74.1 | 75.1 | 75.5 | 10.2 | 20.6 | 81.9 | 56.7 | 60.0 | 57.3 |
|  |  | *Ours* | 68.0 | **80.4** | 81.8 | 83.9 | 15.1 | **80.4** | 83.6 | 58.5 | 57.8 | 67.2 |

Table 2: Results for PHD finetuned on GLUE. The metrics are $F_1$ score for QQP and MRPC, Matthew's correlation for COLA, Spearman's $\rho$ for STS-B, and accuracy for the remaining datasets. Bold values indicate the best model in category (noisy/clean), while underscored values indicate the best pixel-based model.

## 5.1 Language Understanding

We adapt the commonly used GLUE benchmark (Wang et al., 2018) to gauge our model's understanding of language. We convert GLUE instances into images similar to the process described in §4.1. Given a GLUE instance with sentences $s_1, s_2$ ($s_2$ can be empty), we embed $s_1$ and $s_2$ into an HTML template, introducing a line break between the sentences. We then render the HTML files as images.

We generate two versions of this visual GLUE dataset – clean and noisy. The former is rendered using a single pre-defined font without applying degradations or augmentations, whereas the latter is generated with random fonts and degradations. Fig 6 presents a sample of each of the two dataset versions. While the first version allows us to measure PHD's understanding of language in "sterile" settings, we can use the second version to estimate the robustness of the model to noise common to historical scans.

## 5.2 Historical Question Answering

QA applied to historical datasets can be immensely valuable and useful for historians (Borenstein et al., 2023a). Therefore, we assess PHD's potential for assisting historians with this important NLP task. We finetune the model on two novel datasets. The first is an adaptation of the classical SQuAD-v2 dataset (Rajpurkar et al., 2016), while the second is a genuine historical QA dataset.

**SQuAD Dataset** We formulate SQuAD-v2 as a patch classification task, as illustrated in Fig 11 in App C. Given a SQuAD instance with question $q$, context $c$ and answer $a$ that is a span in $c$, we render $c$ as an image, $I$ (Fig 11a). Then, each patch of $I$ is labelled with 1 if it contains a part of $a$ or 0 otherwise. This generates a binary label mask $M$ for $I$, which our model tries to predict (Fig 11b). If any degradations or augmentations are later applied to $I$, we ensure that $M$ is affected accordingly. Finally, similarly to Lee et al. (2022), we concatenate to $I$ a rendering of $q$ and crop the resulting image to the appropriate input size (Fig 11c).

Generating the binary mask $M$ is not straightforward, as we do not know where $a$ is located inside the generated image $I$. For this purpose, we first use Tesseract to OCR $I$ and generate $\hat{c}$. Next, we use fuzzy string matching to search for $a$ within $\hat{c}$. If a match $\hat{a} \in \hat{c}$ is found, we use Tesseract to find the pixel coordinates of $\hat{a}$ within $I$. We then map the pixel coordinates to patch coordinates and label all the patches containing $\hat{a}$ with 1. In about 15% of the cases, Tesseract fails to OCR $I$ properly, and $\hat{a}$ cannot be found in $\hat{c}$, resulting in a higher proportion of SQuAD samples without an answer compared to the text-based version.

As with GLUE, we generate two versions of visual SQuAD, which we use to evaluate PHD's performance in both sterile and historical settings.

**Historical QA Dataset** Finally, we finetune PHD for a real historical QA task. For this, we use the English dataset scraped from the website of the *Runaways Slaves in Britain* project, a searchable database of over 800 newspaper adverts printed between 1700 and 1780 placed by enslavers who wanted to capture enslaved people who had self-liberated (Newman et al., 2019). Each ad was manually transcribed and annotated with more than 50 different attributes, such as the described gender

and age, what clothes the enslaved person wore, and their physical description.

Following Borenstein et al. (2023a), we convert this dataset to match the SQuAD format: given an ad and an annotated attribute, we define the transcribed ad as the context $c$, the attribute as the answer $a$, and manually compose an appropriate question $q$. We process the resulting dataset similarly to how SQuAD is processed, with one key difference: instead of rendering the transcribed ad $c$ as an image, we use the original ad scan. Therefore, we also do not introduce any noise to the images. See Figure 7 for an example instance. We reserve 20% of the dataset for testing.

### 5.3 Training Procedure

Similar to BERT, PHD is finetuned for downstream tasks by replacing the decoder with a suitable head. Tab 4 in App A.1 details the hyperparameters used to train PHD on the different GLUE tasks. We use the standard GLUE metrics to evaluate our model. Since GLUE is designed for models of modern English, we use this benchmark to evaluate a checkpoint of our model obtained after training on the artificial modern scans, but before training on the real historical scans. The same checkpoint is also used to evaluate PHD on SQuAD. Conversely, we use the final model checkpoint (after introducing the historical data) to finetune on the historical QA dataset: First, we train the model on the noisy SQuAD and subsequently finetune it on the *Runaways* dataset (see App A.1 for training details).

To evaluate our model's performance on the QA datasets, we employ various metrics. The primary metrics include binary accuracy, which indicates whether the model agrees with the ground truth regarding the presence of an answer in the context. Additionally, we utilise patch-based accuracy, which measures the ratio of overlapping answer patches between the ground truth mask $M$ and the predicted mask $\hat{M}$, averaged over all the dataset instances for which an answer exists. Finally, we measure the number of times a predicted answer and the ground truth overlap by at least a single patch. We balance the test sets to contain an equal number of examples with and without an answer.

### 5.4 Results

**Baselines** We compare PHD's performance on GLUE to a variety of strong baselines, covering both OCR-free and OCR-based methods. First, we use CLIP with a ViT-L/14 image encoder in the lin-

| Task | Model | Noise / Image | Binary acc | Patch acc | One Overlap |
|---|---|---|---|---|---|
| S | BERT | ✗/ ✗ | 72.3 | 47.3 | 53.9 |
| | *Ours* | ✗/ ✓ | 60.3 | 16.4 | 42.2 |
| | *Ours* | ✓/ ✓ | 61.7 | 14.4 | 41.2 |
| R | BERT | - / ✗ | 78.3 | 52.0 | 55.8 |
| | *Ours* | - / ✓ | 74.7 | 20.0 | 48.8 |

Table 3: Results for PHD finetuned on our visual SQuAD (S) and the *Runaways Slaves* (R) datasets.

ear probe setting, which was shown to be effective in a range of settings that require a joint understanding of image and text—including rendered SST-2 (Radford et al., 2021). While we only train a linear model on the extracted CLIP features, compared to full finetuning in PHD, CLIP is about $5\times$ the size with ~427M parameters and has been trained longer on more data. Second, we finetune Donut (§2.2), which has ~200M parameters and is the closest and strongest OCR-free alternative to PHD. Moreover, we finetune BERT and PIXEL on the OCR output of Tesseract. Both BERT and PIXEL are comparable in size and compute budget to PHD. Although BERT has been shown to be overall more effective on standard GLUE than PIXEL, PIXEL is more robust to orthographic noise (Rust et al., 2023). Finally, to obtain an empirical upper limit to our model, we finetune BERT and PIXEL on a standard, not-OCRed version of GLUE. Likewise, for the QA tasks, we compare PHD to BERT trained on a non-OCRed version of the datasets (the *Runaways* dataset was manually transcribed). We describe all baseline setups in App B.

**GLUE** Tab 2 summarises the performance of PHD on GLUE. Our model demonstrates noteworthy results, achieving scores of above 80 for five out of the nine GLUE tasks. These results serve as evidence of our model's language understanding capabilities. Although our model falls short when compared to text-based BERT by 13 absolute points on average, it achieves competitive results compared to the OCR-then-finetune baselines. Moreover, PHD outperforms other pixel-based models by more than 10 absolute points on average, highlighting the efficacy of our methodology.

**Question Answering** According to Tab 3, our model achieves above guess-level accuracies on these highly challenging tasks, further strengthening the indications that PHD was able to obtain impressive language comprehension skills. Although the binary accuracy on SQuAD is low, hovering

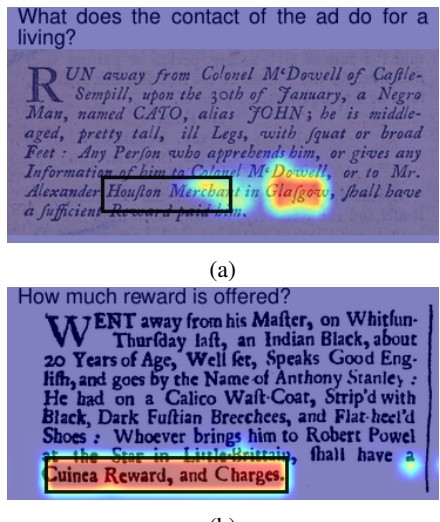

(a)

(b)

Figure 8: Saliency maps of PHD fine-tuned on the *Runaways Slaves in Britain* dataset. Ground truth label in a grey box. The figures were cropped in post-processing.

around 60% compared to the 72% of BERT, the relatively high "At least one overlap" score of above 40 indicates that PHD has gained the ability to locate the answer within the scan correctly. Furthermore, PHD displays impressive robustness to noise, with only a marginal decline in performance observed between the clean and noisy versions of the SQuAD dataset, indicating its potential in handling the highly noisy historical domain. The model's performance on the *Runaways Slaves* dataset is particularly noteworthy, reaching a binary accuracy score of nearly 75% compared to BERT's 78%, demonstrating the usefulness of the model in application to historically-oriented NLP tasks. We believe that the higher metrics reported for this dataset compared to the standard SQuAD might stem from the fact that *Runaways Slaves in Britain* contains repeated questions (with different contexts), which might render the task more trackable for our model.

**Saliency Maps** Our patch-based QA approach can also produce visual saliency maps, allowing for a more fine-grained interpretation of model predictions and capabilities (Das et al., 2017). Fig 8 presents two such saliency maps produced by applying the model to test samples from the *Runaways Slaves in Britain* dataset, including a failure case (Fig 8a) and a successful prediction (Fig 8b). More examples can be found in Fig 15 in App C.

## 6 Conclusion

In this study, we introduce PHD, an OCR-free language encoder specifically designed for analysing historical documents at the pixel level. We present a novel pretraining method involving a combination of synthetic scans that closely resemble historical documents, as well as real historical newspapers published in the Caribbeans during the 18th and 19th centuries. Through our experiments, we observe that PHD exhibits high proficiency in reconstructing masked image patches, and provide evidence of our model's noteworthy language understanding capabilities. Notably, we successfully apply our model to a historical QA task, achieving a binary accuracy score of nearly 75%, highlighting its usefulness in this domain. Finally, we note that better evaluation methods are needed to further drive progress in this domain.

## Acknowledgements

This research was partially funded by a DFF Sapere Aude research leader grant under grant agreement No 0171-00034B, the Danish-Israeli Study Foundation in Memory of Josef and Regine Nachemsohn, the Novo Nordisk Foundation (grant NNF 20SA0066568), as well as by a research grant (VIL53122) from VILLUM FONDEN. The research was also supported by the Pioneer Centre for AI, DNRF grant number P1.

## Limitations

We see several limitations regarding our work. First, we focus on the English language only, a high-resource language with strong OCR systems developed for it. By doing so, we neglect low-resource languages for which our model can potentially be more impactful.

On the same note, we opted to pretrain our model on a single (albeit diverse) historical corpus of newspapers, and its robustness in handling other historical sources is yet to be proven. To address this limitation, we plan to extend our historical corpora in future research endeavours. Expanding the range of the historical training data would not only alleviate this concern but also tackle another limitation; while our model was designed for historical document analysis, most of its pretraining corpora consist of modern texts due to the insufficient availability of large historical datasets.

We also see limitations in the evaluation of PHD. As mentioned in Section 4.4, it is unclear how to empirically quantify the quality of the model's reconstruction of masked image regions, thus necessitating reliance on qualitative evaluation. This qualitative approach may result in a suboptimal model for downstream tasks. Furthermore, the evaluation tasks used to assess our model's language understanding capabilities are limited in their scope. Considering our emphasis on historical language modelling, it is worth noting that the evaluation datasets predominantly cater to models trained on modern language. We rely on a single historical dataset to evaluate our model's performance.

Lastly, due to limited computational resources, we were constrained to training a relatively small-scale model for a limited amount of steps, potentially impeding its ability to develop the capabilities needed to address this challenging task. Insufficient computational capacity also hindered us from conducting comprehensive hyperparameter searches for the downstream tasks, restricting our ability to optimize the model's performance to its full potential. This, perhaps, could enhance our performance metrics and allow PHD to achieve more competitive results on GLUE and higher absolute numbers on SQuAD.

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

## A    Reproducibility

### A.1    Training

**Pretraining**    We pretrain PHD for 1M steps on with the artificial dataset using a batch size of $176$ (the maximal batch size that fits our system) using AdamW optimizer (Kingma and Ba, 2014; Loshchilov and Hutter, 2017) with a linear warm-up over the first $50$k steps to a peak learning rate of $1.5e-4$ and a cosine decay to a minimum learning rate of $1e-5$. We then train PHD for additional 100k steps with the real historical scans using the same hyperparameters but without warm-up. Pretraining took 10 days on $2 \times 80$GB Nvidia A100 GPUs.

**GLUE**    Table 4 contains the hyperparameters used to finetune PHD on the GLUE benchmark. We did not run a comprehensive hyperparameter search due to compute limitations; these settings were manually selected based on a small number of preliminary runs.

**SQuAD**    To finetune PHD on SQuAD, we used a learning rate of $6.75e-6$, batch size of $128$, dropout probability of $0.0$ and weight decay of $1e-5$. We train the model for $50\,000$ steps.

**Runaways Slaves in Britain**    To finetune PHD on the *Runaways Slaves in Britain* dataset, first trained the model on SQuAD using the hyperparameters mentioned above. Then, we finetuned the resulting model for an additional 1000 steps on the *Runaways Slaves in Britain*. The only hyperparameter we changed between the two runs is the dropout probability, which we increased to $0.2$.

### A.2    Dataset Generation

**List of dataset augmentations**    To generate the synthetic dataset described in Section 4.1, we applied the following transformations to the rendered images: text bleed-through effect; addition of random horizontal and lines; salt and pepper noise; Gaussian blurring; water stains effect; "holes-in-image" effect; colour jitters on image background; and random rotations.

**Converting the Caribbean Newspapers dataset into $368 \times 368$ scans**    We convert full newspaper pages into a collection of $368 \times 368$ pixels using the following process. First, we extract the layout of the page using the Python package Eynollah.[8]

---

[8] https://github.com/qurator-spk/eynollah

| Parameter | MNLI | QQP | QNLI | SST-2 | COLA | STS-B | MRPC | RTE | WNLI |
|---|---|---|---|---|---|---|---|---|---|
| Classification-head-pooling | | | | | Mean | | | | |
| Optimizer | | | | | AdamW | | | | |
| Adam $\beta$ | | | | | (0.9, 0.999) | | | | |
| Adam $\epsilon$ | | | | | $1e{-}8$ | | | | |
| Weight decay | | | | | $1e{-}5$ | | | | |
| Learning rate | | | | | $5e{-}2$ | | | | |
| Learning rate warmup steps | | | | | 100 | | | | |
| Learning rate schedule | | | | | Cosine annealing | | | | |
| Batch size | 172 | 172 | 128 | 128 | 128 | 128 | 172 | 172 | 172 |
| Max steps | | | | | 10 000 | | | | |
| Early stopping | | | | | ✓ | | | | |
| Eval interval (steps/epoch) | 500 | 500 | 500 | 500 | 100 | 100 | 100 | 250 | 100 |
| Dropout probability | | | | | 0.0 | | | | |

Table 4: The hyperparameters used to train PHD on GLUE tasks.

This package provides the location of every paragraph on the page, as well as their reading order. As newspapers tend to be multi-columned, we "linearise" the page into a single-column document. We crop each paragraph and resize it such that its width equals 368 pixels. We then concatenate all the resized paragraphs with respect to their reading order to generate a long, single-column document with a width of 368 pixels. Finally, we use a sliding window approach to split the linear page into $368 \times 368$ crops, applying a stride of 128 pixels. We reserve 5% of newspaper issues for validation, using the rest for training. See Fig 10 in App C for dataset examples.

## B  Historical GLUE Baselines

For all baselines below, we compute and average scores over 5 random initializations.

**OCR + BERT/PIXEL**  For each GLUE task, we first generate 5 epochs of noisy training data and run Tesseract on it to obtain noisy text datasets. Similarly, however without oversampling, we obtain noisy versions of our fixed validation sets. We then finetune BERT-base and PIXEL-base in the same way as Rust et al. (2023), with one main difference: the noisy OCR output prevents us from separating the first and second sentence in sentence-level tasks. Therefore we treat each sentence pair as a single sequence and leave it for the models to identify sentence boundaries itself, similar to how PHD has to identify sentence boundaries in the images. We use the codebase and training setup from Rust et al. (2023).[9]

**CLIP**  We run linear probing on CLIP using an adaptation of OpenAI's official codebase.[10] We first extract image features from the ViT-L/14 CLIP model and then train a logistic regression model with L-BFGS solver for all classification tasks and an ordinary least squares linear regression model for the regression tasks (only STS-B).

**Donut**  We finetune Donut-base using an adaptation of ClovaAI's official codebase.[11] We frame each of the GLUE tasks as image-to-text tasks: the model receives the (noisy) input image and is trained to produce an output text sequence such as `<s_glue><s_class><positive/></s_class>`. In this example, taken from SST-2, the `< X >` tags are new vocabulary items added to Donut and the label is an added vocabulary item for the positive sentiment class. All classification tasks in GLUE can be represented in this way. For STS-B, where the label is a floating point value denoting the similarity score between two sentences, we follow Raffel et al. (2020) to round and convert the floats into strings.[12] We finetune with batch size 32 and learning rate between $1e{-}5$ and $3e{-}5$ for a maximum of 30 epochs or 15 000 steps on images resized to a resolution of $320 \times 320$ pixels.

**OCR-free BERT/PIXEL**  For GLUE, we take results reported in (Rust et al., 2021). For SQuAD, we take a BERT model finetuned on SQuAD-v2,[13]

---

[9] https://github.com/xplip/pixel

[10] https://github.com/openai/CLIP#linear-probe-evaluation

[11] https://github.com/clovaai/donut

[12] Code example in https://github.com/google-research/text-to-text-transfer-transformer/blob/main/t5/data/preprocessors.py#L816-L855

[13] from https://huggingface.co/deepset/bert-base-cased-squad2.

and evaluate it on the validation set of SQuAD-v2, after being balanced for the existence of an answer. For the *Runaways Slaves in Britain* dataset, we finetune a BERT-base-cased model[14] on a manually transcribed version of the dataset. We use the default SQuAD-v2 hyperparameters reported in the official Huggingface repository for training on SQuAD-v2.[15] We then evaluate the model on a balanced test set, containing 20% of the ads.

## C    Additional Material

**Figure 9** additional examples from our artificially generated dataset.

**Figure 10** Sample scans from the real historical dataset, as described in Section 4.2.

**Figure 11** The process of generating the *Visual SQuAD* dataset. We first render the context as an image (a), generate a patch-level label mask highlighting the answer (b), add noise and concatenate the question (c).

**Figure 12** Additional examples of PHD's completions over test set samples.

**Figure 13** Dimensionality reduction of embedding calculated by our model on historical scans. We see that scans are clustered based on visual similarity and page structure. However, further investigation is required to determine whether scans are also clustered based on semantic similarity.

**Figure 14** Using PHD for semantic search. Figure 14a and is the target of the search (the concept we are looking for), while Figure 14b and are the retrieved scans.

**Figure 15** Additional examples of PHD's saliency maps for samples from the test set of the *Runaways Slaves in Britain* dataset.

**Figure 16** Examples of shipping ads Newspapers. Newspapers in the Caribbean region routinely reported on passenger and cargo ships porting and departing the islands. These ads are usually well-structured and contain information such as relevant dates, the ship's captain, route, and cargo.

**Figure 17** Input samples for PIXEL. The images are rolled, i.e., the actual input resolution is $16 \times 8464$ pixels. The grid represents the $16 \times 16$ patches that the inputs are broken into.

**Figure 18** An example of a full newspaper page downloaded from the "Caribbean project".

---

[14]from https://huggingface.co/bert-bas e-cased

[15]https://colab.research.google.com/gi thub/huggingface/notebooks/blob/master/e xamples/question_answering.ipynb

ch left the New York contemporary art world and returned to California, taking up a position at California State University, Sacramento where he taught until 2005. Kaltenbach chose to refashion his practice in California, abandoning public conceptual work and instead adopting the persona of a "Regional Artist" with a focus on figurative sculpture and portraiture.
He is also known for work inspired by a found object known as the "Slant Step" which was discovered by William T. Wiley and Bruce Nauman. He has produced drawings, sculptures, films and other work related to the step, most

e party leadership. Hoernle Died to Switzerland: be-engaged actively with the Communist Party there. By the end of 1933 he had made his way to Moscow, which was quickly becoming one of two informal headquarter locations for the exiled Communist Party of Germany
Between December 1933 and November 1940 Hoernle was employed at the International Agriculture Institute in Moscow. Till 1935 he was deputy director of the Department for Central Europe and Scandinavia. Thereafter, for the next five years, he headed up the department. Meanwhile, his progress had not gone unnoticed back in Germany: his German citizenship was revoked in 1938. On a brighter note, there is no indication in the sources of his having been caught up in the Stalinist spy purges which peaked in 1938 and which interrupted or terminated the careers of many other high-profile political refugees from Hitler's version of

Karinthy, the notable Hungarian writer has written his 6 degrees of separation theory at the Central, a full sized photograph of his sitting in the Central is presented in his favourite booth. Gyula Krúdy, the Hungarian journalist and writer wrote his Sinbad themed stories in the Central.
Emboldened by the disastrous defeats of the Ottomans at the hands of Nader and seizing on the pretext that an army of Tatars had violated the sovereignty of Russia by marching along the black sea coast to join Koprulu Pasha against Nader's forces, Russia soon entered into military operations against the Ottoman Empire, eventually capturing Azov. Austria also chose this moment to simultaneously join in a war against Istanbul, however they did not share their Russian ally's success on the field suffering a catastrophic defeat at Grocka.

rey milk cap, but is differentiated by the fact L. vietus milk dries grey, while L. glyciosmus milk dries white. It can also be confused with L. cocosiolens, which also smells of coconuts, but L. cocosiolens has a slimy brown or orange cap and is not found among birch.
Lactarius glyciosmus is a common mushroom and is found under broad-leaved trees, particularly birch-often inside of sphagnum moss. It can be found between late summer and autumn. It grows in soil individually or in scattered groups. It can be found in North America and Europe, New Zealand, Svalbard, Japan, and China.

sionally blogs such as Arcade, a humanities site published by Stanford University. From 2012 to 2016, he hosted a radio show webcast by Alanna Heiss's Clocktower Productions. In autumn 2020, an article he wrote for The Creative Independent was widely disseminated on the internet. Called 19 things I'd tell people contemplating starting a record label (after running one for 19 years) it was a mix of advice, warnings, and personal history gleaned from almost two decades of operating Brassland. It was followed by an appearance on the Third Story podcast.
Sickman's war service took him to Tokyo during the occupation of Japan where he served as one of the "Monuments Men" under General Douglas MacArthur's

terminated by the All England club in 1981 in order for The Championships, Wimbledon to be held. Since then the club has been nomadic, moving to Osterley and Greenford before settling in Acton and playing their matches at Wasps FC's Twyford Avenue Sports Ground. By 2012, the club had downsized to running only one team.
A number of players for the New Zealand national rugby union team have played for London New Zealand including Doug Rollerson, Terry Morrison and Paul Sapsford. In recognition of their history, the club have been granted privileges from both the Rugby Football Union and the New Zealand Rugby. They are the only rugby team aside of New Zealand national representative teams that wears the silver fern as their crest and the RFU exempted them from the overseas player quotas, prior to its abolition. The club have also taken part in a number of New Zealand government

aving been estranged from her father's family for most of her life, Andrea is intrigued. But what exactly is the Bancroft's involvement with "Genesis," a mysterious person working to destabilize the geopolitical balance at the risk of millions of lives? In a series of devastating coincidences, Andrea and Belknap come together and must form an uneasy alliance if they are to uncover the truth behind "Genesis"—before it is too late.
Girls' BMX was part of the cycling at the 2010 Summer Youth Olympics program. The event consisted of a seeding round, then elimination rounds where after three races the top 4

... swimmers have so far achieved qualifying standards in the following events (up to a maximum of 2 swimmers in each event at the Olympic Qualifying Time (OQT), and potentially 1 at the Olympic Selection Time (OST)):
Venezuela has entered one athlete into the table tennis competition at the Games. Gremlins Arvelo secured the Olympic spot in the women's singles by virtue of her top six finish at the 2016 Latin American Qualification Tournament in Santiago, Chile.

thian fronts before he and the corps were transferred to the Italian Front in early 1916, participating in the Trentino Offensive. He had a mixed record as a corps commander. His commanding officer General Svetozar Borević, who had rated him as not suitable for a higher command back in the Carpathians, totally changed his assessment of him after they served together in Italy.
In early 1917 he returned to the Eastern Front. Initially given command of the X Corps, half a year later Křitek succeeded Karl Tersztyánszky von Nádas as commander of the 3rd Army. Meanwhile he had been promoted to Generaloberst

d in 2008. In March 2012, he was inducted into the North-West Frontier Province (NWFP) provincial cabinet of Chief Minister Ameer Haider Khan Hoti and was appointed as Provincial Minister of NWFP for Industries and Commerce. He was re-elected to the Provincial Assembly of Khyber Pakhtunkhwa as an independent candidate from Constituency PK-65 (D.I. Khan-II) in 2013 Pakistani general election. He received 25,921 votes and defeated Tariq Rahim Kundi, a candidate of PPP.

al A ("1935" [1936]). "Contribução ao conhecimento dos ofídios do Brasil. VII. Novos gêneros e espécies de Colubrídeos opisthoglyphos". Mémorias do Instituto Butantan 9: 203–207. (Calamodon paucidens, new species, p. 204). (in Portuguese).
Donovaly is a village in the Banská Bystrica Region of central Slovakia. Being

... 1945 Kleindeutscheck) is a village in the administrative district of Gmina Olecko, within Olecko County, Warmian-Masurian Voivodeship, in northern Poland. It lies approximately north-east of Olecko and east of the regional capital Olsztyn.
Kiš' compositions have been performed by many ensembles such as Asko Ensemble, Maarten Altena Ensemble, Zagreb Philharmonic Orchestra, Netherlands Wind Ensemble and The Croatian Television Symphonic Orchestra. With Tomislav Oliver, Kiš composed the music for Kraljevi bogova, a ballet choreographed by Pascal Touzeau, performed for the first time in Croatian National Theatre in Zagreb in 2015, as a part

Im Sung-jae has always been in love with Eun-hee, but the past threatens to tear them apart. Since his father's death, Sung-jae was ironically rescued and raised as a son by the real killer, Cha Seok-goo.
A West Indian is a native or inhabitant of the West Indies (the Antilles and the Lucayan Archipelago). For more than 100 years the words West Indian specifically described natives of the West Indies, but by 1661 Europeans had begun to use it also to describe the descendants of European colonists who stayed in the West Indies. Some West Indian people reserve this term for citizens or natives of the British West Indies.
British Guiana (now Guyana) competed at the 1948 Summer Olympics in London, England. Four competitors, all men, took part in seven events in three sports. It was the first time that the nation competed at the Olympic Games.
David Andrew McIntosh Parra (, born 17 February 1973) is a retired Venezuelan footballer who played as a centre back. McIntosh also has been capped for the Venezuela national team in two Copa América editions by Eduardo Borrero and José Omar Pastoriza as coach.

r and director Brad Bird, among others. Bird was the first to use the A113 Easter egg, on a car license plate in an animated segment entitled Family Dog in a 1987 episode of the television series Amazing Stories.
The Incredibles – Room number in Syndrome's lair (not seen, only mentioned by Mirage). Also, the prison level, where Mr. Incredible is held is "Level A1" in Cell #13: A1 & 13 and as, Elastigirl looks to the computer, in the "Level A1" diagram, the higher column is labeled "13".
Cars – The number of the freight train that almost crashes into Lightning McQueen while he is first on his way to Radiator Springs. It is also Mater's license plate in both the film and the related short film, Mater and the Ghost Light.

ing Tyhurst and Chiddingstone Cobham. Henry Streatfeild bought Bore Place in 1759. Upon acquiring Bore Place estate was divided in two, with one tenant farmer occupying the main house (South Bore Place) and another living in North Bore Place. Henry himself chose to live at High Street House in Chiddingstone, later known as Chiddingstone Castle, which he had inherited from his father in 1747.
Henry married Lady Anne Sidney, the illegitimate daughter of Jocelyn Sidney of Penshurst Place on 25 September 1752, at Enfield. On Sir Jocelyn's death, Henry could potentially have inherited the Penshurst Estate as the 7th Earl left no legitimate heir and on his death-bed wrote a will leaving everything to his 14-year-old

other organisations for improved pay and working conditions, and in the 1970s it achieved equalisation of welfare and social security with industrial workers, and for workers not to be laid off without just cause.
Employment in the sector gradually declined, and by 1987, it had only 348,621 members. In 1988, it merged with the Italian Federation of Sugar, Food Industry and Tobacco Workers, to form the Italian Federation of Agroindustrial

Figure 9: Samples of our artificially generated dataset, and compare to Figure 10.

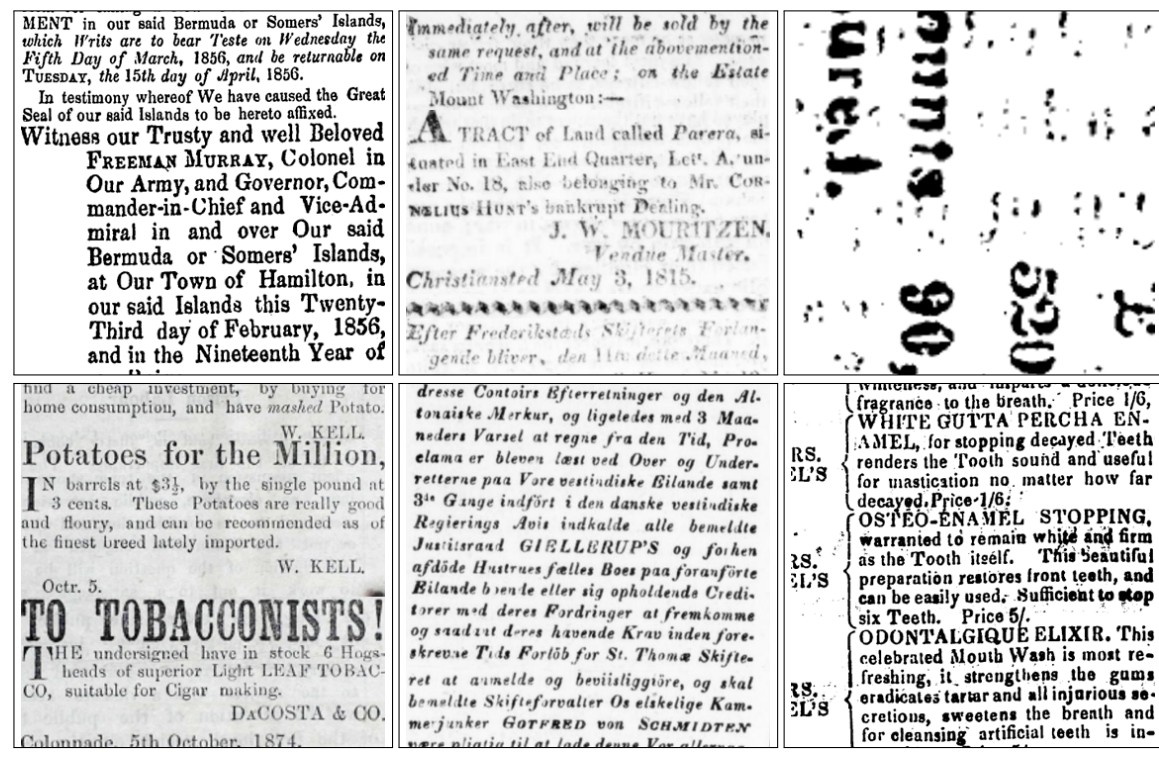

Figure 10: Sample scans from the real historical dataset.

| (a) Rendering context $c$ as an image $I$. | (b) Generating a label mask $M$. | (c) Adding $q$ and degradations. |

Figure 11: Process of generating the *Visual SQuAD* dataset. We first render the context as an image (a), generate a patch-level label mask highlighting the answer (b), add noise and concatenate the question (c).

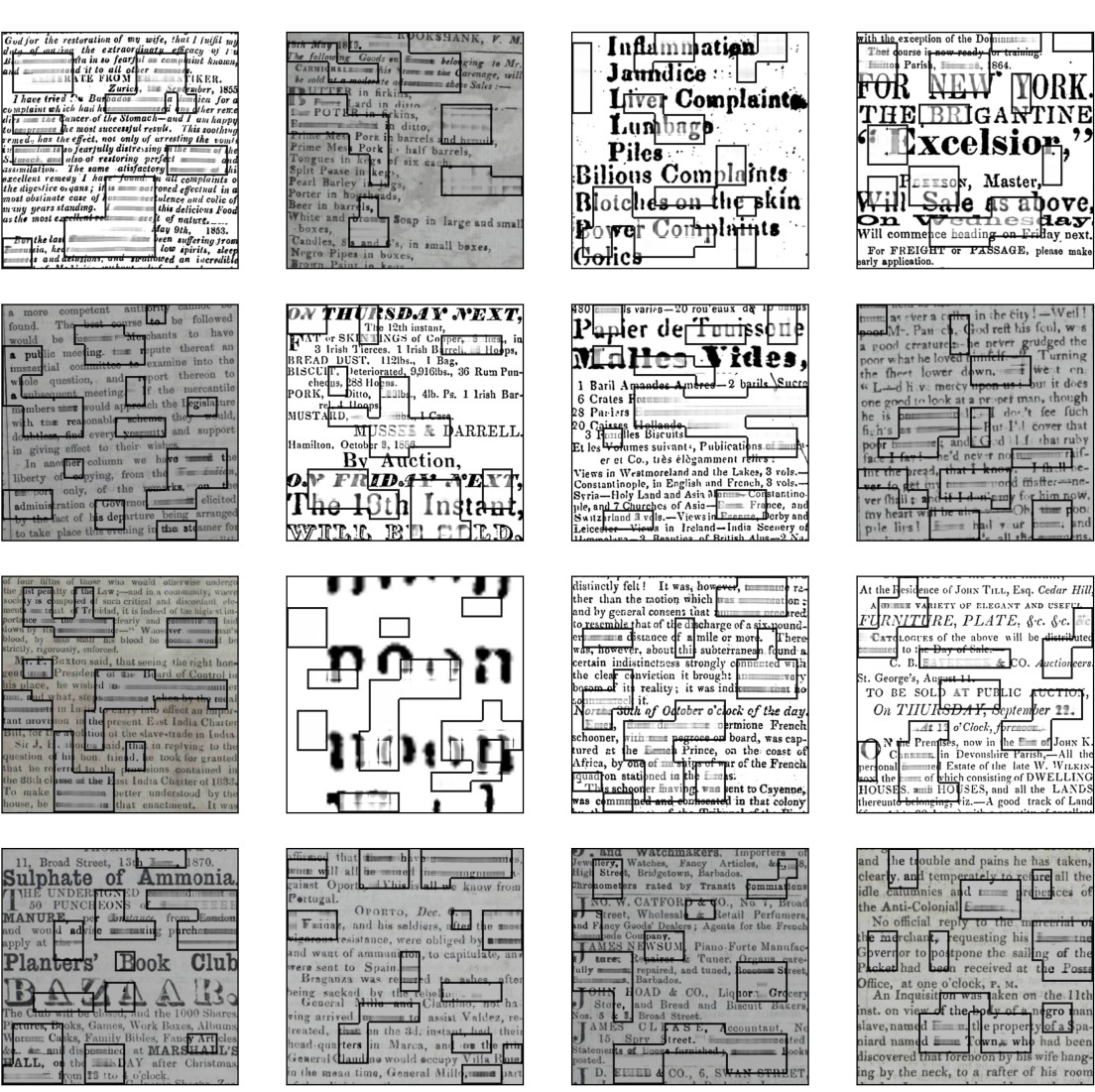

Figure 12: Additional examples of PHD's completions.

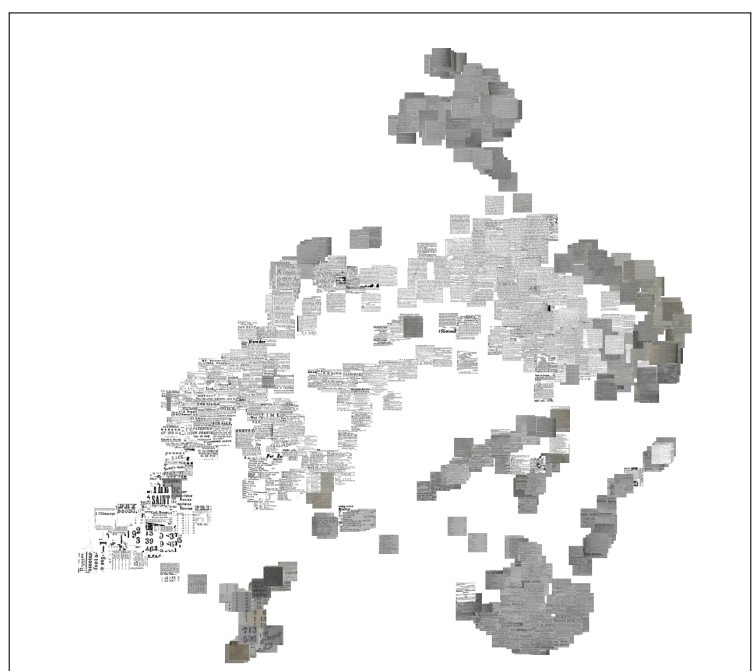

Figure 13: Dimensionality reduction of embedding calculated by our model on historical scans.

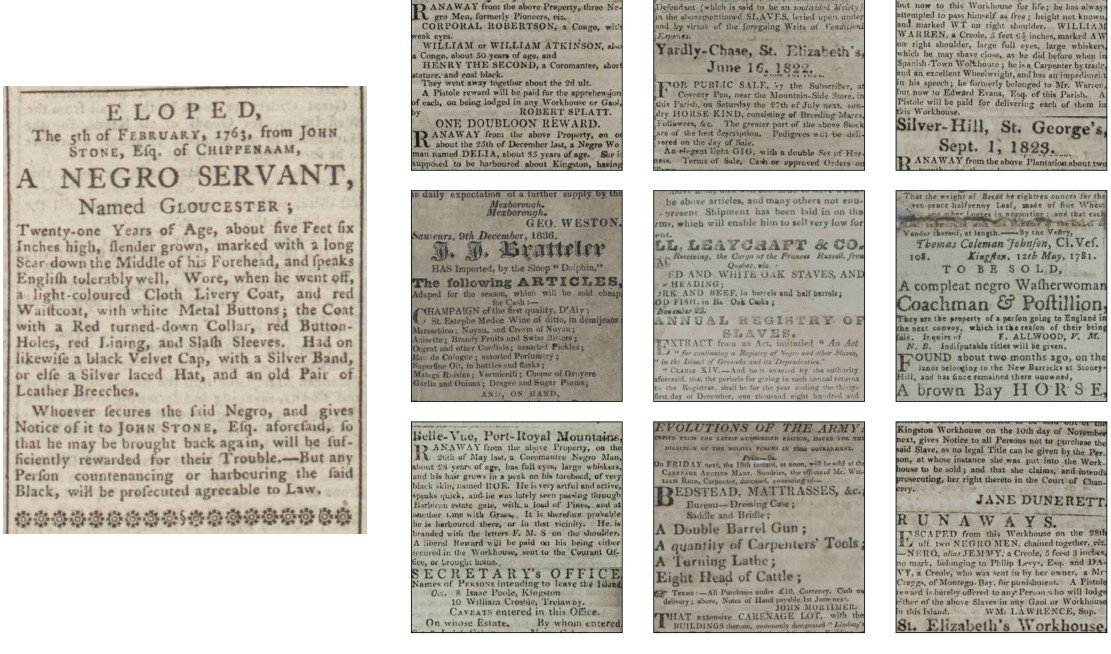

(a) Semantic search target.

(b) Retrieved scans.

Figure 14: Semantic search using our model. (a) is the target of the search, and (b) are scans retrieved from the newspaper corpus.

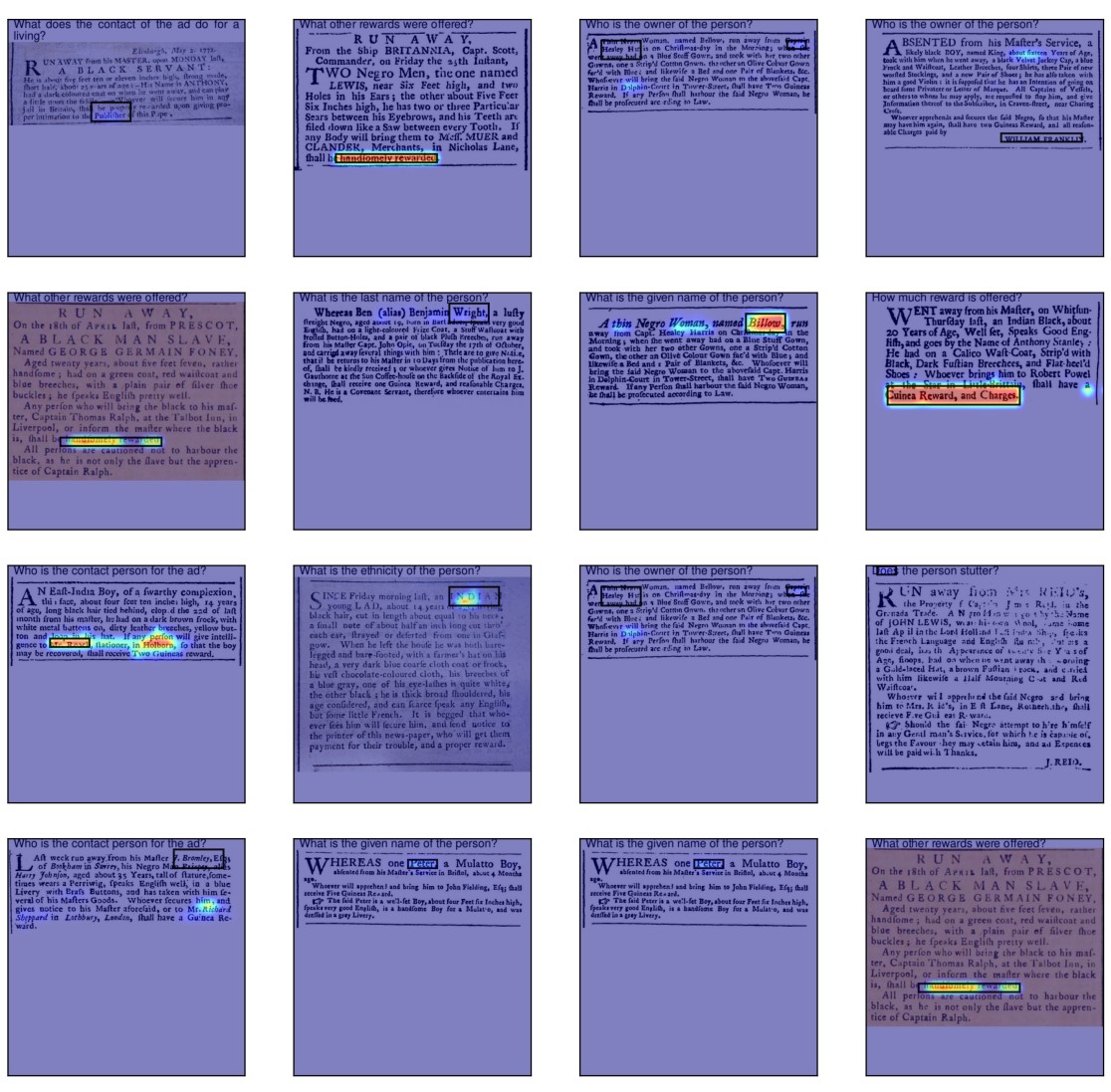

Figure 15: Additional examples of PHD's saliency maps for samples from the test set of the *Runaways Slaves in Britain* dataset.

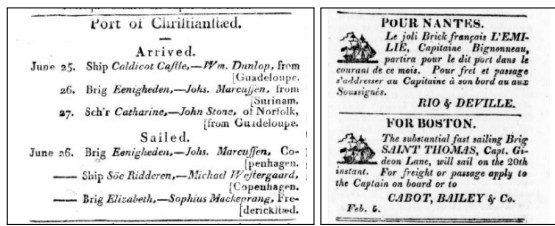

Figure 16: Shipping ads samples. Newspapers in the Caribbean region routinely reported on passenger and cargo ships porting and departing the islands. These ads are usually well-structured and contain information such as relevant dates, the ship's captain, route, and cargo.

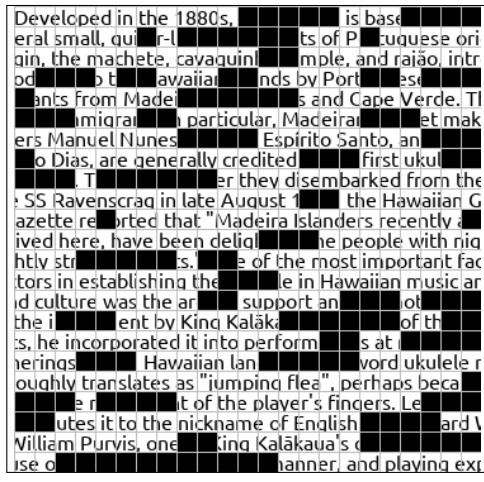

(a) PIXEL's input.

(b) PIXEL's masking.

Figure 17: Input samples for PIXEL. The images are rolled, i.e., the actual input resolution is $16 \times 8464$ pixels. The grid represents the $16 \times 16$ patches that the inputs are broken into.

# THE
# Royal Gazette.
### BERMUDA COMMERCIAL AND GENERAL ADVERTISER AND RECORDER.

No. 24.—Vol. XXXIX.     STATE SUPER VIAS ANTIQUAS.     24s. per Ann

### *Hamilton, Bermuda, Tuesday, June 19, 1866.*

---

**Commissariat, Bermuda,**
HAMILTON, 11TH JUNE, 1866.
TENDERS, in Duplicate. will be received by the Deputy Commissary General, at his Office in *Hamilton,* until Noon of

**SATURDAY,**
The 23rd June,

From Persons willing to supply such Quantities of

## HOPS,

As may be required for Service of the Commissariat Bakeries. between the 1st July, 1866 and 31st March, 1867. Payment for the same to be made Quarterly. Further information can be obtained on application at the COMMISSARIAT OFFICE at *St. Georges.*

**T. W. GOLDIE,**
D. C. G.

[Hamilton papers insert twice.]

---

Articles Adapted to the wants of all
Classes of Society,
CAN BE OBTAINED AT REDUCED RATES
On application at the
**St. Georges, General Store.**

*In addition to other recent Arrivals,*
**THE UNDERSIGNED**
ARE RECEIVING
Per ' Minnie Ha Ha,' ' Forest Fairy,'
' Star of the East,' &c.
The following :—
LAUNDRESSES' IRONS,
Shoemakers' TOOLS,
Bull's Eye LANTERNS
Watering POTS   Toilet CANS
Baths & Bath BOTTLES
Tin Tea KETTLES   Cast Iron DO., (tinned)
Galvanized BUCKETS
Galvanized Round and Oval TUBS
Tea POTS   Table BELLS   Toast FORKS
Milk PANS   Coffee MILLS   Sugar SCOOPS
SCALES and WEIGHTS, ½ oz. to 4lb., to 28lbs. to 56lb. and to 200 lbs.
Also, Spring BALANCES
Quadrant DITTO, &c.,   Coffee MILLS
Round, Oval and Square Bake PANS
Japan'd CANS   Roasting JACKS
Long Spout Oil FEEDERS
Milk SKIMMERS,   SPITTOONS
Enamelled SAUCEPANS
Pocket, Table, Dessert, Oyster and Carving KNIVES,   Steels. SCISSORS, &c.,
Glass PAPER   Emery CLOTH
*Sets Ladies Garden TOOLS*
BROOMS and HANDLES
Stock, Banister and Shoe BRUSHES
—ALSO—
Breakfast, Dinner and Tea SETS
Toilet SETS, &c.,   150 dozen BASINS—suited to Military and Naval Messes.
Cut and Prest GLASS   WINES   TUMBLERS
PRESERVES   Cruet BOTTLES   SALTS
Sugar BASINS   Butter DISHES
Milk EWERS,   &c.,   &c.
☞ *N.B. Harnesses, &c., neatly Made and Repaired.*
**OXBORROW & HUGHES.**
St. Georges, June, 9th 1866.

---

**For Sale,**
Per Recent Importations,
*And per ELIZA BARSS,*
Just from New York,
BBLS. Thin Mess PORK
Ditto Pilot BREAD, small cakes
Ditto fine Yellow Corn MEAL
Boxes CHOCOLATE   Boxes HERRINGS
HOPS   Condensed MILK
Choice BUTTER and CHEESE,
&c.,   &c.,   &c.,
**Green GINGER,**
Boxes FLORIDA WATER
Bbls Choice SUGAR, &c.,   &c.,   &c.
Pure KEROSENE. as harmless as Mr. Anybody's, Warranted, &c., &c., &c.
**B. E. DICKINSON.**
Hamilton, June 12. 1866.—2

---

**Just Arrived,**
PLATED water PITCHERS
Cake BASKETS
Bread BASKETS   Pickle STANDS
Card BASKETS   Spoon HOLDERS
Napkin RINGS, &c., &c.
ALSO,
A Fine Assortment of
**Mourning Brooches, Ear**
Rings, and Silver Thimbles,
AT
**CHILD & GAULTS,**
Reid Street, Hamilton.
June 12, 1866.

Hamilton papers insert four times only.

---

BERMUDA, *Alias* }
SOMERS' ISLANDS. }
*By His Excellency HARRY ST. GEORGE ORD, Companion of the Most Honorable Order of the Bath, Brevet-Colonel in the Royal Engineers, Governor, Commander in-Chief, Vice Admiral and Ordinary, in and over these Islands, &c., &c., &c.*
WHEREAS *MARY FRANCES PITCHER* has prayed for Administration, with Will annexed, on the Estate of ABSALOM CLARKSON PITCHER, late of St. David's Island, in St. Georges Parish in these Islands, Stonemason, Deceased.
This is therefore to give Notice, that if any Person or Persons can shew any just Cause why the said Administration should not be granted unto the said MARY FRANCES PITCHER, he, she, or they are to file his, her, or their Caveat in writing, in the Secretary's Office of these Islands within Fifteen days from the publication hereof. otherwise the said Administration will be granted accordingly.
**MILES GERALD KEON,**
Col. Secretary.
Dated at the Secretary's Office,
this 7th day of June, 1866. }

---

**For Sale,**
BY "ELIZA BARSS,"
AND IN STORE.
BARRELS New T. M. PORK,
Ditto ditto Packet Mess BEEF
Barrels Pilot and Navy BREAD
Barrels FLOUR and CORN MEAL
Bags White and Yellow CORN
Bags BRAN, 5 bushels each
Barrels Brown and White SUGAR
Boxes Honey Dew TOBACCO, 12's
Bristol's SARSAPARILLA,
SOAP and STARCH   HAMS and BACON
Adamantine and Tallow CANDLES
Puns. Demerara RUM,
&c.,   &c ,   &c.,
**B. W. WALKER.**
Hamilton, June 12, 1866 —2

---

**O. C. DUNSCOMBE**
Offers for Sale,
**Ex barque Eliza Barss,**
*FROM NEW YORK,*
**10 Barrels TAR.**
Hamilton, June 12, 1866.

---

**THE SUBSCRIBER**
**HAS RECEIVED,**
His usual supply of
**SUMMER GOODS.**
**FROM LONDON,**
*per Mail Steamer via Halifax,*
Which he offers at a small advance for *Cash* at his Residence.
**FOSTER L. BONNELL.**
Riddells Bay, June 4th, 1866 —

---

**TEAS and COFFEE.**
HALF Chests Congou TEA,
Half Ditto Souchong DITTO,
Half Ditto Oolong DITTO,
Half Ditto best Hyson DITTO,
Boxes   do.   do.   DITTO,
Mocha COFFEE,
Ceylon DITTO,
Java DITTO.
*Wholesale or Retail,*
By
**GOSLING BROTHERS,**
Hamilton and St. Georges.
October 23, 1865.

---

**J. A. Frith,**
**PHOTOGRAPHER,**
*ST. GEORGES,*
Late Calle de las Enramadas, N 13 Santiago de Cuba.

**Cartes de Visite, Vignetts,**
(Spanish Royal Privilege) Double Cards, or the same persons in two positions on the same picture—Porcelain or Albatypes—Ferrotypes for Lockets—Ambrotypes.
PRICES—Half dozen Cards. 10s ; one dozen double Pictures Ditto 12s.; ditto 20s.
Frames of different sizes and prices. Albums.
Ambrotypes with Case from 3s ; to $5.
Hours for Photographing from 10 to 4
Cloudy weather makes no difference in securing a good picture.
January 16. 1866.—6m

---

**An Apprentice Wanted**
to the *TAILORING TRADE.*
Apply to
**T. KERRISK.**
Reid Street, Hamilton, }
April 16th, 1866. }

---

**Mechanics' Industrial**
## EXHIBITION,
In aid of Completing the Association's Hall.
*Under the distinguished Patronage of*
HIS EXCELLENCY THE GOVERNOR
AND MRS ORD.
Wednesday 27th, Thursday 28th, and Friday 29th June, on the Property of MRS. KENNEDY, known as
**" Richmond Grounds."**

THE PUBLIC are respectfully informed that HIS EXCELLENCY THE GOVERNOR has kindly consented to open the Exhibition on

## WEDNESDAY

27th June,
And that it will be continued the two following days viz. :—the 28th and 29th.
As the Mechanics' Hall, when completed, is to used chiefly for educational purposes, which is hoped to prove advantageous to the Country at large. the Committee most earnestly solicit aid by way of donations from every class of the Public. Contributions of every possible description will be thankfully received at the Store of MR. HARNETT, *Hamilton,* and placed in the deposit room, which has been secured for the purpose through the courtesy of MRS. DR. HARVEY.
During the time of the Exhibition every care will be taken to promote the comfort of the visitors. Booths will be erected for shelter from the sun, and Refreshments in great variety will be prepared for the occasion, which, in conjunction with the display of Goods. both local and foreign—hitherto unequalled in these Islands—and other arrangements now being made, it is hoped that all who may visit the grounds will be pleasantly entertained.
*By authority of the Committee,*
**C. W. GAUNTLETT,**
Secretary.
May 29, 1866.

---

## ICE.
**THE SUBSCRIBERS**
Are Now Receiving
**THEIR USUAL SUPPLY OF**
## ICE.
Which they will commence to Issue
On the 1st June.
TERMS will be made known on application at their Store.
**GOSLING BROS.**
Hamilton, May 30. 1866.

---

**VICTORIA HOTEL,**
**Front Street, Hamilton.**
THE above HOTEL has just been reopened by its former Proprietress, Mrs. C. SLATER, who mindful of former liberal patronage extended to her, and feeling grateful for all past favors again ventures to solicit the support of her Friends and the Public generally in the revived Establishment, which she trusts will continue to deserve and receive the countenance of the community
BREAKFASTS. LUNCHEONS. DINNERS. TEAS, &c, provided at the shortest notice, and on Moderate Terms.
The House is now ready to receive Boarders.
Hamilton. April 27th, 1866.

---

## SODA WATER.
**Bottled Soda Water,**
Of a Superior quality can be supplied in any quantity from the Medical Hall, St. Georges.
**W. R. HIGINBOTHOM.**
St. Georges. May 1st, 1866.—2m.

---

**EXPERIMENT HILL,**
FOR RENT.
This Commodious **Mansion** in the Town of Hamilton, will accommodate a very large Family ; or two families may conveniently occupy it.
From the upper Story it commands a beautiful and an extensive view. It has just been put in good order for a tenant, and immediate possession can be given.
Apply at Miss Wood's Seminary, Hamilton, May 1. 1866.

---

## NOTICE.
The Subscriber offers for Rent the **Warehouse,** on Queen Street lately occupied by himself.
**WM. J. COX.**
Hamilton, May 22, 1865.

---

BERMUDA, *Alias.* }
SOMERS' ISLANDS. }
*By His Excellency HARRY ST. GEORGE ORD, Companion of the Most Honourable Order of the Bath, Brevet-Colonel in the Royal Engineers, Governor, Commander-in-Chief and Vice-Admiral in and over these Islands, &c., &c., &c.*
[L.S.]
H. St. George Ord, Governor and Commander-in-Chief.

### A Proclamation.

WHEREAS information has reached ME. THE GOVERNOR AND COMMANDER-IN-CHIEF aforesaid, that **CHOLERA** has appeared at the Ports of HALIFAX and NEW YORK:—I DO THEREFORE, by virtue of the power and authority in me vested. by an Act of the Legislature of these Islands, entitled, "An Act to consolidate and amend the Quarantine Laws," and by and with the advice and Consent of HER MAJESTY'S COUNCIL for these Islands, hereby issue this MY PROCLAMATION, and do hereby make known that the said Ports of Halifax and New York are Infected Places within the meaning of the said Act.—And I do hereby strictly charge and Command all Pilots going on board or taking charge of any Vessel arriving at these Islands from either of the aforesaid Ports forthwith to conduct the same to some one of the Quarantine Stations prescribed by the above named Act. there to remain until she shall be visited by the advice and Consent of HER MAJESTY'S COUNCIL for these Islands, hereby issue this MY PROCLAMATION, and do hereby give such orders and directions as the circumstances of each case may justify and to his said office may pertain.
Given under My Hand and the Great Seal of these Islands this Second day of May, 1866, and in the twenty ninth year of Her Majesty's Reign.
*By His Excellency's Command,*
**MILES GERALD KEON,**
Colonial Secretary.

GOD SAVE THE QUEEN!

---

BERMUDA, *Alias,* }
SOMERS' ISLANDS. }
*By His Excellency HARRY ST. GEORGE ORD, Companion of the Most Honourable Order of the Bath, Brevet-Colonel in the Royal Engineers, Governor, Commander-in-Chief, and Vice-Admiral in and over these Islands, &c., &c., &c.*
[L. S.]
H. St. George Ord, Governor and Commander-in-Chief.

### A Proclamation.

WHEREAS information has reached ME, THE GOVERNOR AND COMMANDER-IN-CHIEF aforesaid, that **CHOLERA** has appeared at GUADALOUPE, one of the French West India Islands :—I DO THEREFORE, by virtue of the power and authority in me vested, by an Act of the Legislature of these Islands, entitled "An Act to Consolidate and Amend the Quarantine Laws," and by and with the advice and consent of HER MAJESTY'S COUNCIL for these Islands, hereby issue this MY PROCLAMATION, and do hereby make known that the said Island of Guadaloupe is an infected place within the meaning of the said Act:—And I do hereby strictly Charge and Command all PILOTS going on board or taking charge of any Vessel arriving at these Islands from the aforesaid place, forthwith to conduct the same to some one of the Quarantine Stations prescribed by the above named Act, there to remain until she shall thereupon give such orders and directions as the circumstances of each case may justify, and to his said office may pertain.
Given under My Hand and the Great Seal of these Islands this twenty-third day of December, 1865, and in the two. twenty-ninth year of Her Majesty's Reign.
*By His Excellency's Command,*
**MILES GERALD KEON,**
Colonial Secretary.

GOD SAVE THE QUEEN!

---

# Bermuda.
*Colonial Secretary's Office,*
JUNE 1. 1866.
THE following ACT, which was passed by the Legislature of Bermuda in the month of *September,* 1855, having been laid before Her Majesty in Council, together with a letter to the Lord President of the Council from the Right Honble Edward Cardwell, one of Her Majesty's Principal Secretaries of State, recommending that the said Act should be left to its operation, Her Majesty was ther upon pleased by and with the advice of her Privy Council to approve the said recommendation.
**MILES GERALD KEON,**
Colonial Secretary.

16.—An Act further to amend the Act No. 4 of 1850, relating to Liquor Shops.

