# OpenReview forum: "PHD: Pixel-Based Language Modeling of Historical Documents"
_EMNLP/2023/Conference — EMNLP 2023 Main_

### Official Review · Reviewer_NET7 · 2023-08-05

**Soundness:** 3

**Excitement:**

4: Strong: This paper deepens the understanding of some phenomenon or lowers the barriers to an existing research direction.

**Missing References:**

- Lewis, David D. et al. “Building a test collection for complex document information processing.” Proceedings of the 29th annual international ACM SIGIR conference on Research and development in information retrieval (2006): n. pag.

- Mathew, Minesh et al. “DocVQA: A Dataset for VQA on Document Images.” 2021 IEEE Winter Conference on Applications of Computer Vision (WACV) (2020): 2199-2208.

**Paper Topic And Main Contributions:**

This paper introduces PHD, a pixel-based language model for analyzing historical documents without using OCR. The main contributions are:

1. Proposes a novel method to generate synthetic scans that resemble historical documents for pretraining. This helps address the scarcity of large historical scan datasets.

2. Pretrains PHD on a combination of synthetic scans and real historical newspapers from the 18th-19th centuries.

3. Evaluates PHD on image reconstruction, clustering, language understanding tasks like GLUE, and question answering on both SQuAD and a real historical QA dataset.

4. Provides evidence that PHD can effectively understand language and has potential for assisting with NLP tasks involving historical documents.

5. Releases the datasets, models, and code to facilitate future research.

Overall, this paper explores using recent advances in pixel-based language modeling to process historical scans directly at the pixel level. This allows bypassing the OCR stage which can introduce noise when applied to historical documents. The proposed pretraining methodology and evaluations demonstrate the promise of this approach for historical document analysis.

**Questions For The Authors:**

The previous studies utilized datasets such as IIT-CDIP and DocVQA for pre-training and evaluation. Can you discuss why you did not consider them?

**Reasons To Accept:**

This paper explores an interesting new direction and provides a thorough evaluation of the proposed techniques. Releasing the datasets and models could catalyze more work in this area.
1. Well-written paper that clearly explains the motivation, proposed approach, experiments, results, and limitations.
2. Novel application of pixel-based language models to historical document analysis, bypassing the need for OCR. This is an interesting new direction for processing historical texts.
3. Releases new datasets, models, and code to facilitate research in this area. The historical QA dataset created from real newspaper ads could be valuable for the community.

**Reasons To Reject:**

1. Most of the pretraining data is modern text, not historical. More diverse historical data could help the model better adapt to that domain. Previous work, such as Donut, DiT, Dessurt, and LayoutLM (v1, v2, v3), pre-trained their models on IIT-CDIP. IIT-CDIP is a large-scale scanned document corpus used for pre-training language models.

2. The evaluation tasks, apart from one historical QA dataset, predominantly involve modern text. More historical evaluation data could better assess performance.

3. There are also some document understanding benchmarks, such as DocVQA (also used in Donut, DiT, Dessurt, and LayoutLM v1, v2, v3), which can be used to evaluate the question-answering performance of models.

4. As the paper mentions, evaluating the pixel-based completions is challenging. More robust quantitative evaluation methods are needed.

5. OCR techniques continue to improve over time. At some point, OCR quality may be sufficient to apply standard NLP pipelines to historical texts without needing to bypass OCR.

**Reproducibility:**

3: Could reproduce the results with some difficulty. The settings of parameters are underspecified or subjectively determined; the training/evaluation data are not widely available.

**Reviewer Confidence:**

4: Quite sure. I tried to check the important points carefully. It's unlikely, though conceivable, that I missed something that should affect my ratings.

---

> ### Author Rebuttal · Authors · 2023-08-27
>
> We thank the reviewer for their insightful and detailed feedback. We are pleased that you believe that our work is interesting and novel and that the paper is well-written. We hope that we can address some of your concerns:
>
> > *“Most of the pretraining data is modern text, not historical…”*
>
> A: Reviewer **Uoyc** raised this concern as well, which we also note in the manuscript (e.g., lines 221). We will better clarify the following key points in the camera-ready version:
> 1. Historical corpora are either too small to train a strong LLM successfully or cannot be easily obtained. Therefore, using modern texts is a compromise we must make.
> 2. We claim that the model can learn general, transferable language understanding skills from contemporary English texts, while their adaptation to the historical domain can be learned from the smaller historical corpus. This pretraining approach is commonly (and successfully) used in many analogue problems. For example, to train a model on a low-resource language, it is beneficial to also train the model on another, high-resource language.
>
>
> > *“...previous work, such as Donut, DiT, Dessurt, and LayoutLM, pre-trained their models on IIT-CDIP…”*
>
> A: We considered adding IIT-CDIP to our pretraining corpora. However, we dismissed this idea as the dataset's domain is narrow and dissimilar to historical texts (it consists of documents from a state's lawsuit against the tobacco industry in the 1990s). We will add this consideration to a footnote in the camera ready.
>
>
> > *“The evaluation tasks, apart from one historical QA dataset, predominantly involve modern text. More historical evaluation data could better assess performance.”*
>
> A: This is an important limitation of our work, which we also acknowledged in the limitation section of the paper (line 605). Following your comment and Reviewer **Uoyc**'s request, we will augment the end of section 4.4 (Clustering) by conducting experiments involving semantic search over historical texts. Specifically, we will use PHD to encode a large set of historical scans **S** and a target scan **t** containing a specific topic of interest. We will then use cosine similarity to find scans in **S** that share the same topic as **t**. We hope this additional benchmark will give more insights into PHD’s ability to understand historical language.
>
>
>
> > *“There are also some document understanding benchmarks, such as DocVQA…, which can be used to evaluate the question-answering performance of models.”*
>
> A: This is an interesting suggestion. While these are modern datasets and, as such, not the focus of our paper, they can still provide some insights into the capabilities of PHD. We will add an appendix where we analyse the performance of PHD on these datasets.
>
>
> > *“OCR techniques continue to improve over time. At some point, OCR quality may be sufficient to apply standard NLP pipelines to historical texts without needing to bypass OCR.”*
>
> A: This is an interesting and valid perspective. Nevertheless, we anticipate that it will require many years before OCR models reach a level of quality that guarantees freedom from errors. Furthermore, as Nguyen et al. [Survey of Post-OCR Processing Approaches. ACM Computing Surveys, 2021] point out, OCR engines will face difficulties with historical texts even if they perform well on modern texts, because *“they still lack adequate training data composed of past documents”*. In the meantime, OCR models could potentially propagate errors into downstream tasks and pipelines.

---

### Official Review · Reviewer_Uoyc · 2023-08-05

**Soundness:** 4

**Excitement:**

4: Strong: This paper deepens the understanding of some phenomenon or lowers the barriers to an existing research direction.

**Paper Topic And Main Contributions:**

The paper proposes a method for historical document reconstruction based on PIXEL, a language model that’s unique in that it deals in visual, rather than token-based, representations of language.  The main contribution of this paper is a language model that follows the general design of PIXEL but is trained on a synthetically generated corpus of historical documents.

**Reasons To Accept:**

The method proposed is well-argued and the limitations are clearly discussed.  The evaluation is robust, evaluating both the model’s ability to resolve corruption of both synthetic and actual historical documents, as well as measuring the model’s language understanding against standard benchmarks.

**Reasons To Reject:**

This reviewer sees no core or overwhelming reason to reject this study; the contribution of the authors’ can be characterized as adapting an existing method to a novel domain, which may not be compelling to some, but it is an interesting and thorough study of the applicability of those ideas to said domain.

The cluster analysis may be superfluous — it’s not clear what the authors hoped to understand by performing that study if not the effectiveness of the encoder at providing a deep semantic representation of historical documents, however the authors noted that they only evaluated the visual similarity of similarly encoded documents.

One limitation perhaps not adequately discussed is that much of the synthetic training corpus consists of contemporary English that’s rendered in the same font and style as a historical document, and also much of the language understanding evaluation is based on contemporary English as well; depending on the time period in which the documents of interest are written, shifts in style, the meaning of words, etc., could limit the applicability of the model’s language understanding — however, evaluation on actual historical documents is performed, so this limitation is to a degree quantified.

**Reproducibility:**

4: Could mostly reproduce the results, but there may be some variation because of sample variance or minor variations in their interpretation of the protocol or method.

**Reviewer Confidence:**

4: Quite sure. I tried to check the important points carefully. It's unlikely, though conceivable, that I missed something that should affect my ratings.

---

> ### Author Rebuttal · Authors · 2023-08-27
>
> Thank you for reviewing our manuscript and for your useful suggestions. We are glad that you enjoyed our work and believe that our method is well-argued and our evaluation robust.
>
>
> > *“The cluster analysis may be superfluous — it’s not clear what the authors hoped to understand by performing that study…”*
>
> A: This is a valid criticism. In this experiment, we hoped to gain initial insights into the usefulness of PHD’s encoding in search and retrieve tasks. However, we acknowledge the experiment’s shortcomings. Therefore, for the camera-ready version, we will augment this section by conducting comprehensive experiments involving semantic search. Specifically, we will use PHD to encode a large set of scans **S** and a target scan **t** containing a specific topic of interest. We will then use cosine similarity to find scans in **S** that share the same topic as **t**.
>
>
> > *“...the synthetic training corpus consists of contemporary English that’s rendered in the same font and style as a historical document... [this] could limit the applicability of the model’s language understanding…”*
>
> A: While we note this concern in the manuscript (e.g., lines 221), we agree that the paper can benefit from further developing this discussion. We will do so for the camera-ready version. We will better clarify the following key points:
> 1. Historical corpora are either too small to train a strong LLM successfully or cannot be easily obtained. Therefore, using modern texts is a compromise we must make.
> 2. We claim that the model can learn general, transferable language understanding skills from contemporary English texts, while their adaptation to the historical domain can be learned from the smaller historical corpus. This pretraining approach is commonly (and successfully) used in many analogue problems. For example, to train a model on a low-resource language, it is beneficial to also train the model on another, high-resource language.
>
> > *“also, much of the language understanding evaluation is based on contemporary English as well;...”*
>
> A: This is an important limitation of our work, which we also acknowledged in the limitation section of the paper (line 605). While we leave the development of a historical variant of GLUE to future work, we will include in the camera-ready version a benchmark of semantic search over historical texts (as mentioned above). We hope this additional benchmark will give more insights into PHD’s ability to understand historical language.
>
>
> > *“The contribution of the authors’ can be characterized as adapting an existing method to a novel domain, which may not be compelling to some…”*
>
> A: While we acknowledge and are pleased that you do not share this view, we want to stress that our work goes beyond simply applying an existing model to a new domain. Rather, several methodological innovations had to be developed in creating PHD, such as the pretraining method, dataset generation and processing, and evaluation schemes.

---

### Official Review · Reviewer_nvph · 2023-08-10

**Soundness:** 4

**Excitement:**

4: Strong: This paper deepens the understanding of some phenomenon or lowers the barriers to an existing research direction.

**Paper Topic And Main Contributions:**

This paper proposes a new modeling approach called PHD with pixel-based language modeling for scanned (historical) documents. Instead of modeling with text tokens (typically generated by OCR engines), the method takes in patches of document images (pixels) and trains a language model on top of it. A similar masked language modeling objective is used to pre-train such models on a combination of synthetic and real data; the objective is to “black out” specific image patches and the goal is to recover the original text image. Empirical results on both general NLU tasks like GLUE and historical document QA tasks show that PHD can achieve good and sometimes even matching performances compared to text-based language models like BERT.


**Questions For The Authors:**

See above.

**Reasons To Accept:**

This is a pretty novel work and has the potential to inspire many follow up work for understanding scanned (historical) documents. It has the following strengths:
- Simplicity: the image-based modeling approach is a generalization of the PIXEL model in the previous work. No special technique is involved in pre-processing the images – simply dividing any input image to patches and PHD can treat each individual patch as a token and model them like a “language”.
- PHD has a clever design that uses the same representation space in both the input and the output. Compared to previous work like TrOCR that tries to convert input images to text output, in PHD, the input and output are both image patches. It eliminates some difficulties like aligning image and text (which often involves much post-host processing and needs special loss like CTC to handle this).
- Another interesting aspect in PHD is how it can be used for “text-only” classification or QA tasks like GLUE or SQUAD. The authors firstly render the text in images, and the PHD model generates a prediction based on the image.


**Reasons To Reject:**

I don’t see any reasons why this paper should be rejected, while the paper can benefit with a discussion of the following:
- For almost all the examples shown in the paper, the cropping seems to be perfect – that is, no cross text line cropping. It would be helpful to include examples (e.g., in fig 13) to show what happens if the answer patch lies at the boundary of the image patch and is half cropped.
- It will be helpful to include a discussion of the efficiency comparison between PHD and BERT. For example, given the same 512 token width, it is unclear whether PHD can encode the same number of actual text compared to BERT.
- The choice of the 16 pixel height for each patch seems to be chosen specifically tuned for the given documents you have. However I can imagine there could be  images with different scanning resolution (e.g., 200 dpi), sticking to 16 as the patch height might not lead to optimal results. It would be better to show some results testing the generalizability of the trained models.


**Reproducibility:**

4: Could mostly reproduce the results, but there may be some variation because of sample variance or minor variations in their interpretation of the protocol or method.

**Reviewer Confidence:**

4: Quite sure. I tried to check the important points carefully. It's unlikely, though conceivable, that I missed something that should affect my ratings.

---

> ### Author Rebuttal · Authors · 2023-08-27
>
> Thank you for your insightful feedback and kind words. Your recognition of our work’s merits and novelty is greatly appreciated.
>
>  >*“It would be helpful to include examples … to show what happens if the answer patch lies at the boundary of the image patch and is half cropped”.*
>
> A: This is a good suggestion. We will supplement the appendix with such examples, as well as other edge cases. We note, however, that the accuracy of the page-layout analysis tool that we used (Eynollah) is very high, so such edge cases are uncommon.
>
>
> > *“... Include a discussion of the efficiency comparison between PHD and BERT… it is unclear whether PHD can encode the same number of actual text compared to BERT.”*
>
> A: Another good idea, which we will add to the camera-ready. Specifically, we will include an analysis of the performance of PHD as a function of font size and text length.
>
>
> > *“The choice of the 16-pixel height for each patch seems to be chosen specifically for the documents you have…”*
>
> A: We want to clarify that the Vision-transformer architecture we used (like most popular ViT models), is capable of processing only 16x16 pixel patches. Therefore, we didn't select this particular hyperparameter to align with the characteristics of our documents; rather, we had to adhere to it. Additionally, we adjust the resolution of documents, either upscaling or downscaling them, to conform to this 16x16 patch size (refer to line 897 in Appendix A.2). This ensures that PHD can effectively handle documents with varying DPIs. We will better elucidate this point by integrating Appendix A.2 into the main body of the paper.

---

### Meta-Review · Area_Chair_mF1N · 2023-09-19

**Recommendation:** 4

**Metareview:**

The reviewers all agree that the paper proposed a novel and useful approach to analyzing scanned historical documents, building on previous work in the area. The problem of analyzing historical text is of great interest to social scientists and researchers in the humanities, and the proposed method could greatly improve their ability to work with historical text. The use of synthetic training data is clever, and the evaluations are well done. One concern is that the training data is taken from modern, as opposed to historical text. The authors, however, provide a reasonable data availability justification for this point. The potential reasons to reject (modern text, 16 pixel size, confusing cluster analysis, missing comparisons to corpora and BERT) are well responded to by the authors in the rebuttal period.

---

### Decision · Program_Chairs · 2023-10-07

**Decision:**

Accept-Main

**Comment:**

The reviewers all agree that the paper proposed a novel and useful approach to analyzing scanned historical documents, building on previous work in the area. The problem of analyzing historical text is of great interest to social scientists and researchers in the humanities, and the proposed method could greatly improve their ability to work with historical text. The use of synthetic training data is clever, and the evaluations are well done. One concern is that the training data is taken from modern, as opposed to historical text. The authors, however, provide a reasonable data availability justification for this point. The potential reasons to reject (modern text, 16 pixel size, confusing cluster analysis, missing comparisons to corpora and BERT) are well responded to by the authors in the rebuttal period.